# Contribution of Arbuscular Mycorrhizal and Endophytic Fungi to Drought Tolerance in *Araucaria araucana* Seedlings

**DOI:** 10.3390/plants12112116

**Published:** 2023-05-26

**Authors:** Daniel Chávez, Gustavo Rivas, Ángela Machuca, Cledir Santos, Christian Deramond, Ricardo Aroca, Pablo Cornejo

**Affiliations:** 1Departamento de Ciencias y Tecnología Vegetal, Universidad de Concepción, Campus Los Ángeles, Juan Antonio Coloma 0201, Los Ángeles 4440000, Chile; gurivas@udec.cl (G.R.); angmachu@udec.cl (Á.M.); cderamond@udec.cl (C.D.); 2Departamento de Ciencias Químicas y Recursos Naturales, Universidad de La Frontera, Av. Francisco Salazar 01145, Temuco 4811230, Chile; cledir.santos@ufrontera.cl; 3Estación Experimental del Zaidín, CSIC, Profesor Albareda N°1, 18008 Granada, Spain; ricardo.aroca@eez.csic.es; 4Escuela de Agronomía, Facultad de Ciencias Agronómicas y de los Alimentos, Pontificia Universidad Católica de Valparaíso, Quillota 2260000, Chile

**Keywords:** monkey puzzle tree, arbuscular mycorrhizal fungi, endophytic fungi, water stress, plant physiology, bioinoculants

## Abstract

In its natural distribution, *Araucaria araucana* is a plant species usually exposed to extreme environmental constraints such as wind, volcanism, fires, and low rainfall. This plant is subjected to long periods of drought, accentuated by the current climate emergency, causing plant death, especially in its early growth stages. Understanding the benefits that both arbuscular mycorrhizal fungi (AMF) and endophytic fungi (EF) could provide plants under different water regimes would generate inputs to address the above-mentioned issues. Here, the effect of AMF and EF inoculation (individually and combined) on the morphophysiological variables of *A. araucana* seedlings subjected to different water regimes was evaluated. Both the AMF and EF inocula were obtained from *A. araucana* roots growing in natural conditions. The inoculated seedlings were kept for 5 months under standard greenhouse conditions and subsequently subjected to three different irrigation levels for 2 months: 100, 75, and 25% of field capacity (FC). Morphophysiological variables were evaluated over time. Applying AMF and EF + AMF yielded a noticeable survival rate in the most extreme drought conditions (25% FC). Moreover, both the AMF and the EF + AMF treatments promoted an increase in height growth between 6.1 and 16.1%, in the production of aerial biomass between 54.3 and 62.6%, and in root biomass between 42.5 and 65.4%. These treatments also kept the maximum quantum efficiency of PSII (Fv/Fm 0.71 for AMF and 0.64 for EF + AMF) stable, as well as high foliar water content (>60%) and stable CO_2_ assimilation under drought stress. In addition, the EF + AMF treatment at 25% FC increased the total chlorophyll content. In conclusion, using indigenous strains of AMF, alone or in combination with EF, is a beneficial strategy to produce *A. araucana* seedlings with an enhanced ability to tolerate prolonged drought periods, which could be of great relevance for the survival of these native species under the current climate change.

## 1. Introduction

*Araucaria araucana* (Mol.) K. Koch (class Pinopsida, family Araucariaceae, monkey puzzle tree) is a plant species of scientific, ancestral/cultural, and nutritional importance. It is an endemic plant found in the temperate rainforest in northern Patagonia, including southern Chile and southwestern Argentina. Due to its high endemism and longevity, *A. araucana* is a unique genetic resource worldwide [1]. Unfortunately, *Araucaria araucana* has been exposed to both intense logging and extraction of its seeds, in addition to natural events such as wind, volcanism, and fires [2,3].

Regarding the above, prolonged warmer seasons have brought disastrous consequences for this plant species, with a greater recurrence of forest fires that have seriously affected different native species in central–southern Chile [3,4]. The decrease in rainfall has also become an important variable to consider in Chile since, in the last 10 years, drastic climate change has been observed in the country. This phenomenon has evolved and is now recognized as an extreme megadrought [5,6]. Based on the presence of markedly low soil humidity, this conifer is affected in its root growth, especially by the lower accumulation of biomass, which is concomitant with the increase in the mortality rate [7,8,9]. Some species of the genus *Araucaria* (*A. bidwili*, *A. heterophylla*, and *A. cunninghamii)* are highly resistant to drought, possessing mechanisms that involve the production of high levels of abscisic acid (ABA), making them isohydric spent species with efficient stomatal closure [10,11]. However, *A. araucana* presents the typical traits of an ancient species, such as large stomata and low stomatal density, which trigger low conductance and a slow stomatal response to changing environmental conditions [12].

Symbiotic association with soil microorganisms could be a strategy to allow the plant to tolerate even prolonged drought periods [13]. *A. araucana* generates mutualistic associations with microorganisms present in the soil, as do most terrestrial plants, receiving multiple benefits. These microorganisms are mainly represented by arbuscular mycorrhizal fungi (AMF), which develop inside the roots of *A. araucana* as well as in longitudinal fine roots (LFR) and globular short roots (GSR) [14], with abundant hyphae and coils in root cells. This may be ecologically relevant in terms of the importance of this symbiosis in response to soil nutrient deficiencies, mainly P. Additionally, AMF generates stability within plant communities [15], allows better absorption of nutrients and water, and protects the plant from attacks by pathogenic organisms. In exchange, these fungi receive photoassimilates from the plants [11,13]. AMF can promote plant growth traits such as shoot and root biomass and improve the plant’s CO_2_ assimilation rates, which can result in greater plant resistance to some types of stress, such as drought [16,17].

Endophytic fungi (EF) are also found inside the plant tissues of *A. araucana* [18]. As observed for AMF, EF can also improve nutrient absorption and plant development and increase the production of different phytohormones. In this way, EF can promote increased plant growth and resistance to various abiotic stresses [19,20,21,22]. However, recent studies have shown that the benefits of these symbiotic fungi can vary depending on environmental and nutritional conditions, changing from a mutualistic state to parasitism, where the EF would be the most sensitive to those changes [23,24]. Although the application of fungal inoculum from AMF or EF has been studied individually (single-species cultures), based on their potential functional and physiological complementarity, fungal consortia currently play a key role in plant development. Depending on the soil conditions where fungal consortia are established, plants can achieve greater resistance and tolerance to adverse biotic and abiotic factors [25,26].

Understanding the benefits that could be provided by both AMF and EF to plants under different water regimes would generate inputs to address the above-mentioned issues in the case of *A. araucana*. Therefore, to demonstrate the beneficial effect of bioinoculants on the drought tolerance of this plant species, the main aim of this work was to evaluate the effect of AMF and EF inoculation (individually and in combination) on the morphophysiological variables of *A. araucana* seedlings subjected to different water regimes.

## 2. Results

### 2.1. Fungal Root Colonization

The evaluation of *A. araucana* roots after 5 months of fungal inoculation revealed high colonization percentages in the plants inoculated with EF and EF + AMF (Figure 1a). In the case of the EF group, colonization reached 58%, whereas for, the EF + AMF combination, it was ca. 60%. These values were significantly higher than those of plants inoculated only with the AMF consortium, which reached 41% root colonization (Figure 1a).

The roots of *A. araucana* plants inoculated with AMF showed a low number of arbuscules but a high quantity of intraradical mycelia and vesicles in the cortex (Figure 1b(i–vi)). On the other hand, the roots colonized by the EF showed a large number of structures, such as microsclerotia and melanized hyphae (Figure 1b(vii–viii)). It should be noted that the root architecture showed variations among the different treatments (Figure 1c), with a larger number of fine roots in those plants treated with AMF and EF + AMF. In contrast, the control treatment presented heavy, lignified roots and poor secondary root growth (Figure 1c).

In general, all the experimental variables were influenced by both the main sources of variation (fungal inoculation and irrigation) as well as the interactions (Table 1). In detail, variables such as Fv/Fm, RWC, proline, and total chlorophyll stand out as having a highly significant *p* < 0.001 value for the factor interaction, while irrigation and fungal inoculation strongly influenced significant changes in the growth traits (height, DHN, and shoot and root biomass).

### 2.2. Morphological Traits and Mortality Rate

The seedlings inoculated with AMF and the EF + AMF combination at 100 and 75% irrigation presented a significantly greater height than the control group (without application of inocula). In contrast, for 25% irrigation, the AMF presented the greatest height increase, yielding significant differences from all the other treatments (Figure 2a). In the case of the treatment with EF only, no significant differences from the control were observed for any of the irrigation levels.

The largest diameters at the height of the neck of *A. araucana* seedlings were obtained for plants at 100% irrigation, with the AMF and EF + AMF treatments stimulating the greatest increase in diameter, presenting significant differences with the control and EF treatments. No significant differences between the inoculated and control plants were observed at the 25 and 75% irrigation levels (Figure 2b).

The highest aerial biomass production (4.5 g) was obtained by applying AMF and EF + AMF at 100 and 75% irrigation (Figure 2c), presenting significant differences from the control and EF treatments. In contrast, the lowest aerial biomass production (1.23 g) was observed for the control group with 25% irrigation, significantly lower than for the case of the plants inoculated under the same irrigation conditions (Figure 2c). The control and EF groups at 25% irrigation presented the lowest root biomass production (1.23 and 1.3 g, respectively) (Figure 2d). The value of root biomass production observed was significantly lower than plants inoculated with AMF and EF + AMF with the same irrigation level (2.3 and 2.5 g, respectively) (Figure 2d).

The shoot:root ratio showed that EF presented the highest value (2.27) at 25% irrigation, presenting significant differences from all the other treatments for the same irrigation level. No significant differences were observed between the treatments for the 100 and 75% irrigation levels (Figure 2e).

Regardless of the inoculum type applied, all plants, including the control plants, survived in the 100% and 75% irrigation regimes. However, when irrigation decreased to 25% of field capacity, mortality was observed in almost all the inoculation treatments. Only plants inoculated with the EF + AMF consortium presented the lowest mortality rate of 25%. In contrast, plants inoculated with EF reached the highest mortality rate of 87.5%, followed by uninoculated plants and plants inoculated with AMF, which presented 75 and 50% mortality rates, respectively (Table 2).

### 2.3. Maximum Quantum Yield of Photosystem II (Fv/Fm)

After two months of plant growth, the control group of *A. araucana* seedlings and the seedlings inoculated with EF and subjected to 25% irrigation presented the lowest and similar photosynthetic performance, with Fv/Fm ratios of 0.39 and 0.44, respectively (Figure 3a). However, the groups of seedlings inoculated with AMF and EF + AMF presented Fv/Fm values of 0.71 and 0.64, respectively, at the same irrigation level. These values are close to the optimal range of Fv/Fm (100% irrigation) and are significantly higher than those of the control seedlings and those inoculated exclusively with EF (Figure 3a).

### 2.4. Relative Water Content (RWC)

Both the *A. araucana* seedlings inoculated with EF and the control plants at 25% irrigation presented the greatest loss of foliar water content. In both cases, the RWC values were statistically different from the other treatments (Figure 3b). The water loss for the EF group was 25% compared to seedlings inoculated with EF and subjected to 100% irrigation. For the control group, the water loss was 33% (Figure 3b). However, seedlings inoculated with AMF and EF + AMF and subjected to 25% irrigation kept foliar RWCs of 61% and 66%, respectively, close to optimal values (75–80%).

### 2.5. Leaf Proline Determination

The control group and the plants with the EF and EF + AMF treatments at 25% irrigation presented the highest proline concentrations, and the values were significantly higher than the AMF treatments (Figure 3c). Overall, the treatments subjected to 100% and 75% irrigation showed low proline concentrations, with significant differences among the applied treatments (Figure 3c).

### 2.6. Concentration of Photosynthetic Pigments

At 25% irrigation, the total chlorophyll concentration in each control assay and in plants inoculated with EF and AMF showed no significant differences. Additionally, under this irrigation regime, the total chlorophyll concentration in the group of plants inoculated with EF + AMF was significantly higher (9.5 mg cm^2^) (Figure 3d). In contrast, the group inoculated with EF at 100% irrigation presented the lowest chlorophyll concentration compared to the other treatments under the same irrigation condition (Figure 3d). For 75% irrigation, it was noted that the treatments with AMF and EF + AMF obtained the highest chlorophyll content (8.2 and 7.6 mg cm^2^), with the AMF group presenting significant differences with all the other treatments.

### 2.7. CO_2_ Assimilation

Under the 75% irrigation regime, every group of *A. araucana* seedlings showed the best response in terms of CO_2_ assimilation rates (Figure 4). In contrast, the control group at 25% irrigation (32.7 ppm) and the *A. araucana* seedlings inoculated with EF (16.3 ppm) were affected by drought stress, presenting photorespiration.

The groups inoculated with AMF and EF + AMF at 100% and 25% irrigation showed similar CO_2_ assimilation rates (−28.0 and −28.6 ppm; −48.3 and −46.0 ppm, respectively). With both treatments (AMF and EF + AMF), the *A. araucana* seedlings subjected to 25% irrigation showed stable CO_2_ assimilation under drought stress compared to the control and EF groups (Figure 4).

### 2.8. Multivariate Associations

The principal component analysis (PCA) reflected the formation of highly homogeneous groups of experimental variables (Figure 5a), where PC1 explains 45.1% and PC2 23.7% of the total experimental variance. According to the component matrix of PCA (not shown), PC1 was highly and positively influenced by the root biomass, diameter at the height of the neck (DHN), Fv/Fm ratio, RWC, and shoot biomass, whereas PC2 was highly and positively influenced by the pigment contents [Chlorophyll a and b (Chl a and Chl b), total chlorophyll (Chltotal), and carotenoids].

A hierarchical cluster analysis established five homogeneous groups of individuals according to the different treatments studied, highlighting: (i) A group associated with high levels of proline and CO_2_ respiration with plants uninoculated and inoculated with EF at 25% irrigation. (ii) Another group comprised exclusively of AMF-colonized plants at 100% irrigation associated with high values of morphometric and physiological traits (height, shoot biomass, DHN, relative chlorophyll, and RWC). (iii) A third group, including mainly the individuals with the dual inoculation at 100% irrigation, representing high root and shoot values, biomass, RWC, DHN, and Fv/Fm. (iv) A group mainly included individuals of dual-inoculated plants growing at 25% irrigation and highly associated with high pigment contents. (v) A group made up of the AMF-inoculated treatments at 75 and 25% irrigation plus some individuals from other treatments, slightly related with variables such as height, shoot biomass, DHN, relative chlorophyll, and RWC (Figure 5b).

## 3. Discussion

Our study was conducted with *A. araucana* seedlings inoculated with EF, AMF, and an EF + AMF combination. First, plants were grown for 5 months under standard greenhouse conditions. Then, to understand the effect of drought stress on inoculated and uninoculated (control) plants, the *A. araucana* seedlings were subjected to three different irrigation regimes (100, 70, and 25% FC) for two months. The root colonization by the AMF (45%) after 5 months of inoculation was low compared to the EF (58%). This could be due to the colonization capacity of the AMF being subject to multiple environmental, physiological, and phenological factors of the plants [27]. One of these factors is the increase in humidity and the decrease in oxygen in the soil, which inhibit the development of mycorrhizal fungi in the roots [28,29]. This situation could explain the low colonization observed in the roots of *A. araucana* because the seedlings were inoculated in autumn.

Overall, compared to the uninoculated control plants, the different types of fungal inocula affected the height and DHN of plants subjected to different irrigation regimes. According to the results obtained in this work, the 7-month duration of the study (5 months of colonization plus 2 months of water stress) allowed for substantially significant morphophysiological changes in the groups of plants subjected to drought stress. Despite the sluggish growth of *A. araucana*, with 3.10–3.55 cm height growth per year [30], the inoculation with AMF and EF + AMF at 100 and 75% irrigation exhibited the greatest height increases, while the treatment with AMF was the most effective under the greatest water stress (25% irrigation). In this respect, studies by [31] observed a greater tolerance to drought stress in Cupressus atlantica with AMF, which was cultivated in pots under different water regimes, where the inoculation promoted growth and a greater RWC in leaves, agreeing with what was found in this study. Evidence indicates that AMF improves host plant performance against various abiotic stresses such as drought [32,33,34], heat, salinity, metals, and extreme temperatures [35,36]. Li et al. [37] suggested that soil microbes such as AMF often regulate the plant growth response to drought stress, alleviate drought damage by increasing photosynthesis and antioxidant enzyme activities, and reduce malondialdehyde (MDA) levels. AMF colonization can also stimulate various physiological responses to drought stress, including stomatal conductance sensitivity, CO_2_ assimilation, and a decrease in relative water content; furthermore, leaf water potential is likely to be improved by AMF inoculation [38,39]. The water absorption rate of AMF hyphae was 2–7 times higher under drought than that of well-watered hyphae, indicating the importance of AMF hyphae to plants under drought conditions [40,41].

The production of shoot biomass showed a more significant increase in *A. araucana* seedlings inoculated with AMF and EF + AMF and subjected to different irrigation regimes compared to the control group. This is related to the beneficial effect of the association with AMF, which contributes to a greater capture of water and nutrients by plants, stimulating their growth [42]. In contrast, for root biomass at 100 and 75% irrigation, no significant differences from the control plants were noted; however, when water stress increased (25% irrigation), the plants inoculated with AMF and EF + AMF exhibited greater root biomass. In this respect, it was demonstrated that the roots of the control plants presented lignified growth with no secondary root development. However, the roots of the plants inoculated with AMF or EF + AMF presented changes in the morphology of the roots that were finer and branched, which makes the absorption of water and nutrients more efficient [43,44]. In contrast, the effects of an EF consortium on the biomass of seedlings subjected to stress (25% irrigation) were negative, mainly affecting root development (Figure 2d). However, the effects of the EF were positive when inoculated together with the AMF, resulting in significant differences in biomass compared to the control group. These results suggest that the EF did not promote higher root biomass or tolerance to extreme drought stress for *A. araucana*. Previous studies have shown that some EF (e.g., *Penicillium* spp. and *Phialocephala* spp.) are dominant in conifers at root level [45], and such microorganisms are xerotolerant, supporting dry environments. However, it is unknown if this mechanism can be granted to the plant via symbiosis. Studies in this regard have been carried out only for grasses, where xerotolerance has been obtained via symbiosis using EF from the Clavicipitaceous group, fungi that can only colonize grasses and not woody species [46,47]. On the other hand, the shoot:root ratio shows higher index values in EF related to lower root production compared to aerial shoots. This may explain the higher stress level for the plant since a high ratio of this indicator would have a low potential to avoid drought stress [48].

Seedlings inoculated with AMF and the EF + AMF treatment presented higher RWC values under drought stress (25% irrigation) than the control and EF groups (Figure 3b). Similar results have been obtained in a study on Olea europaea, where different irrigation regimes were tested in plants inoculated with AMF. According to the authors, the plants growing under abundant irrigation do not produce significant differences in RWC. However, of the plants subjected to a water deficit, only plants inoculated with AMF significantly improved water absorption [49,50,51]. Furthermore, stable RWC values under drought stress have been found for *Cupressus atlantica*, *Casuarina equisetifolia*, and *Citrus tangerine* [52,53,54]. This is because mycorrhizae can: (a) improve water absorption through the network of hyphae, which extends beyond the root itself; (b) generate morphological changes in plant roots [55]; and (c) stimulate the synthesis of abscisic acid [56].

Studies in *A. araucana* [9] under water restriction showed a reduction in RWC, triggering a high production of free proline in leaves. The results obtained in the present study agree with these data from the literature: the lowest RWC values were obtained even at 25% irrigation, related to a high proline content (Figure 3c). Here, RWC values of 61 and 66% were achieved for inoculated plants subjected to water stress of 25% during irrigation. However, compared with other studies developed for the same plant species without fungal inoculation [9], the RWC values are even more extreme (20–50%).

Regarding photosynthetic pigments, the total chlorophyll content did not show significant differences among the control, EF, and AMF groups under drought stress (25% irrigation). In this case, the effect of the inoculum had no impact on this variable (Figure 3d). However, inoculation of *A. araucana* seedlings with the EF + AMF combination significantly impacted total chlorophyll under drought stress (25% irrigation), obtaining the highest pigment concentrations (Figure 3d). Similar results were found in a study using *Verbascum lychnitis* plants inoculated with EF + AMF [26]. The authors reported that the increase in photosynthetic pigments depended on the type of endophyte that interacted with the AMF [26]. The inoculation of *Nicotiana tabacum* with AMF and endophytic bacteria under drought stress was also previously assessed [57]. Increases in total chlorophyll were observed under drought stress, showing high tolerance of the plant after inoculation [57].

The capacity of CO_2_ assimilation by plants is influenced by different environmental factors, where high temperatures, drought, and salinity can reduce the rate of CO_2_ assimilation [58]. In the present study, the percentage of foliar water found in the control and EF groups under drought stress is correlated with low photosynthetic efficiency and reduced CO_2_ assimilation (Figure 4). In fact, drought stress induces stomatal closure to minimize water loss, also causing a decrease in CO_2_ assimilation [59]. According to [60], when exposed to high concentrations of NaCl, *Robinia pseudoacacia* plants inoculated with *Rizophagus irregularis* remained stable in terms of CO_2_ assimilation. In addition, it has been observed that mycorrhization of *Solanum lycopersicum* plants grown under NaCl stress improved the net rate of CO_2_ assimilation by increasing stomatal conductance and photoprotection of PSII processes [61]. Stress due to salt and drought can have the same effect, resulting in less water entering the plant [62]. The results obtained for *A. araucana* inoculated with AMF are consistent with the data in the literature.

In the present study, the *A. araucana* seedlings inoculated with EF, EF + AMF, and the control presented the highest levels of proline accumulation in the leaf tissue under drought conditions (25% irrigation) (Figure 3c and Figure 5). The group of seedlings inoculated only with EF also presented a high concentration of proline but with a mortality rate similar to or higher than the control plants. This suggests that EF could promote a stress condition in the seedlings that was impossible to monitor in the present study. In this regard, Kia et al. [63] described *Arabidopsis thaliana* and *Hordeum vulgare* inoculated with an EF consortium and subjected to different abiotic stress conditions as being affected by EF that competed for nutrients and water. According to the authors, as a result, trophic dependence was seen to affect the fitness of the plants.

Regarding osmoprotectants, the results from previous studies with plants inoculated with AMF and subjected to drought stress have been contradictory. For instance, an increase in proline levels has been described in inoculated plants compared to non-inoculated plants [64,65]. However, other studies have shown a decrease in proline in plants inoculated with AMF under stress from drought or salinity [66,67]. Here, *A. araucana* seedlings inoculated with AMF showed the lowest proline levels, suggesting that these plants improved tolerance to drought stress. In contrast, plants inoculated with EF + AMF presented a high proline concentration at 25% irrigation. In addition, this group showed the lowest mortality rate and stable physiological traits to confront drought stress.

The results presented in our study suggest that the combination of these microorganisms (AMF and EF) was beneficial for *Araucaria araucana* seedlings since EF stimulates proline synthesis [68] and AMF attenuates the possible adverse effects of endophytes under abiotic stress [25,69,70]. Moreover, fungal inocula can promote high percentages of water in the leaves, notably improving tolerance to drought stress [71]. These results suggest the importance of the combination of symbiotic microorganisms in both the rhizosphere and plant organs in improving tolerance to abiotic stress.

## 4. Materials and Methods

### 4.1. Soil Samples and Plant Material

Soil samples for AMF isolation were collected from the rhizosphere of adult *A. araucana* trees from the Nahuelbuta National Park in the Araucanía Region (PNN: 37°47′00″ S, 72°59′00″ W). Soils presented pH 5.04, P Olsen 8.29 (mg P L^−^^1^), Na 227 (mg kg^−^^1^), K 142 (mg kg^−^^1^), Ca 71 (mg kg^−^^1^), and SOM (12.7%) [15]. Plant samples, such as leaves and roots, were extracted from adult plants of *A. araucana* and transferred to the Fungal Biotechnology Laboratory at the Universidad de Concepción, Los Angeles Campus, to perform the corresponding EF isolations.

### 4.2. AMF Spore Isolation and Identification

Arbuscular mycorrhizal fungus spores were isolated from soils using wet sieving and sucrose density gradient centrifugation. Briefly, 25 g of soil were passed through sieves of 500, 125, 45, and 32 µm and thoroughly washed with distilled water. The last soil portion collected in meshes of 45 and 32 µm was distributed in plastic tubes.

Twenty-five mL of the spore suspensions were transferred into Falcon tubes of 50 mL, and 25 mL of a 70% sucrose aqueous solution were inserted at the bottom of the tubes and centrifuged at 600× *g* for 2 min. After centrifugation, samples were decanted, washed with water, and transferred to Petri dishes. Sorting and quantification took place under the dissection microscope at up to 400-fold magnification.

The number of AMF spores was expressed per 100 g of dry soil. For identification, spores were mounted on microscope slides in polyvinyl alcohol-lactic acid glycerol (PVLG) medium [72,73]. The spores were identified based on morphological characteristics such as spore wall structures, subtending hyphae, and germination structures.

Identification reports [74,75] and institutional collections of original species descriptions were used for all AMFs. These analyses were carried out by Myconativa (https://myconativa.com/, accessed on 2 September 2020). *Acaulospora laevis*, *A. scrobiculata*, *A. punctata*, *Scutellospora calospora*, *Claroideoglomus cloroideum*, *C. etunicatum*, *Funneliformis mosseae*, *Paraglomus occultum*, *Glomus badium*, *G. intrarradices*, and *Gigaspora margarita* were identified and used to prepare the AMF inoculum in trap plants.

### 4.3. Cultivation of Trap Plants

To activate the AMF consortium, *Tajetes patula* [76] seedling plants were used as hosts in trap pots, cultivated in a substrate composed of vermiculite and peat in a 2:1 ratio. After 6 months, root colonization was evaluated by cutting and mixing with the substrate, thus obtaining the active inoculum to be applied to the *A. araucana* seedlings.

### 4.4. EF Isolation

For the isolation of EF, the leaves and roots of *A. araucana* previously collected were used. First, fungal isolation was performed according to Vaz et al. [77]. Next, plant roots were carefully washed to remove all traces of soil. Then, roots were washed with sterile distilled water and subsequently superficially sterilized using 10% sodium hypochlorite for 10 min and 70% ethanol for 1 min.

For the needles, 2% sodium hypochlorite was used for 3 min and 75% ethanol for 30 s. Roots and needles surface-sterilized were washed with abundant sterile distilled water, cut into pieces of 5 mm, and distributed in Petri dishes containing malt extract agar 1%. Samples were incubated at 24 °C for 7 days. Fungal strains were then isolated and purified, obtaining a total of 37 endophytic fungi, which were identified using molecular tools.

The genomic DNA extraction procedure was performed according to [78]. For amplification of the internal transcribed spacer (ITS) region, the primer pairs ITS5/ITS4 were used [79]. The amplification products were purified with the PureLink PCR Purification Kit (Invitrogen, EEUU), following the manufacturer’s instructionsand the sequences were obtained using the Multipurpose DNA Sequencing Platform of the Bioscience Center/UFPE (Recife, Brazil).

Regarding EF identification, 19 out of 37 isolates were identified at the species level and 10 out of 37 at the genus level. In addition, 8 out of 37 isolates were not identified. From the 19 isolates identified at the species level, 4 of them were selected for subsequent trials. *Phialocephala fortinii*, *Penicillium melinii*, *Umbelopsis dimorpha*, and *Preussia cymatomera* were selected and used [80,81,82,83]. Fungal selection was based on the characteristics of species that stimulate plant growth. These species were individually cultured in 250 mL of 1% malt extract liquid medium at 24 °C and under static conditions for 20 days. The mycelium obtained was filtered, mixed with sterile distilled water, and ground for 5 s in a blender. This crushed mycelium was later used as EF inoculum (mix: *P. fortinii*, *P. melinii*, *U. dimorpha*, and *P. cymatomera*).

### 4.5. Inoculation of A. araucana Seedlings and Different Water Regimes

Ninety-six seeds of *A. araucana* were superficially sterilized and germinated in Peat/Perlite 1:1 in the greenhouse of the Universidad de Concepción, Los Angeles Campus (Los Angeles, Chile). Eight months after germination, the *A. araucana* seedlings were transplanted into plastic bags containing a sterilized substrate composed of peat and perlite in a 1:1 ratio. Seedlings were inoculated with the AMF and EF consortia described above.

For the inoculation of *A. araucana*, an aliquot of 25 g of the AMF inoculum consortium (obtained from the trap plants) was directly applied to the seedling roots. For the EF consortium, the *A. araucana* seedlings were inoculated with 10 mL of crushed mycelium (1.6 g of fresh inoculum per plant). Then, the EF consortium was directly injected into the root systems of the plants with a syringe at three different points. Seedlings of *A. araucana* were left for 5 months for root colonization.

The trials included four inoculation treatments (control, EF, AMF, and EF + AMF) with three irrigation levels each (100, 75, and 25%, respectively) and eight replicates per treatment (*n* = 8). The trials were kept under irrigation at field capacity from 1 to 2 times per week for 5 months. Subsequently, the plants were subjected to different irrigation levels: (i) at 100% (field capacity), (ii) at 75% (slight water deficit), and (iii) at 25% (high water deficit), according to Zarik et al. [31].

For the above, a pot with dry soil was weighed, obtaining P1. This pot was then watered until saturated, letting it drain for 24 h to obtain P2. The difference (P2 − P1) corresponded to 100% field capacity. To obtain the volume of water for 75% and 25% treatments, calculations were made as follows: 0.75 × (P2 − P1), and 0.25 × (P2 − P1), respectively. The seedlings were grown, watered twice a week for 2 months, and finally harvested. The total time of the trial was 7 months: 5 months post-inoculation and two months of drought stress.

### 4.6. Fungal Root Colonization

Colonization levels by AMF and EF were determined in four plants after cleaning the roots in a 2.5% KOH aqueous solution (*w/v*) and staining with 0.05% trypan blue [84]. According to Giovanetti and Mosse, the presence of AMF structures within the roots was observed at 40–100× in a gridded Petri dish [85]. Root endophytes were quantified by the magnified intersection method [86] using a Motic BA 310 microscope. Melanized sclerotia and hyphae were observed for EF expressed in %.

### 4.7. Maximum Quantum Efficiency of Photosystem II (Fv/Fm)

At the end of the eighth week of water stress, the maximum quantum efficiency of photosystem II (Fv/Fm) of *A. araucana* leaves was measured. A Hansatech pocket model PEA chlorophyll fluorimeter was used, and 8 plants per treatment were measured. Measurements were made on dark-adapted leaves using leaf clips for 20 min. The calculation was made in the instrument using the chlorophyll fluorescence ratio (Fv/Fm), which reflects the maximum quantum efficiency of the PSII [87].

### 4.8. Relative Water Content (RWC)

Leaf relative water content (RWC) [88] was measured using the first four leaves of three plants per treatment (0.5 g). To obtain the fresh leaf mass, leaves were cut and immediately weighed (Fresh Weight, FW). Leaves were then immersed in deionized water in a Petri dish and incubated overnight in the dark, below 8 ºC. To obtain turgid weight (TW), leaves were re-weighed. To determine the dry weight (DW), leaves were placed in an oven at 60 °C for 2 days and weighed again. Finally, RWC was calculated as [(FW − DW) × (TW − DW) − 1] × 100 [89].

### 4.9. Concentration of Photosynthetic Pigments

The photosynthetic pigments were determined according to Aroca et al. [89]. Three leaf discs of 5 mm each were extracted from the third leaf, and immersed in 1 mL of acetone (80% *v/v*). The sample was subsequently heated at 80 °C for 10 min to extract the pigments. The absorbances of the extracts were measured at 470, 646.8, and 663.2 nm using a TU-1810 split beam spectrophotometer. The extinction coefficients and the equations used to obtain pigment concentrations were reported by Lichtenthaler [90]. Each measurement was taken from three plants per treatment.

### 4.10. Proline Content

The free proline concentration was determined using 0.5 g of fresh leaf tissue and spectrophotometrically assayed at 530 nm, as described by Bates et al. [91]. Each measurement was taken from three plants per treatment.

### 4.11. CO_2_ Assimilation

To determine the assimilation of CO_2_ by plants, the Targas-1 portable photosynthesis and gas measurement system was used. Measurements were taken in the morning (from 9 a.m. to 12 p.m.), and three plants per treatment were used. For the CO_2_ measurements of the needles of *A. araucana*, one of the accessories of the device was adapted, the SRC-2 soil respiration chamber (in CPY mode), to obtain assimilation of CO_2_ (negative values) and respiration (positive values). The measurement conditions were 1000 µmol m^−^^2^ s^−^^1^, with a temperature of 22 °C, and a CO_2_ concentration of 400 µmol.

### 4.12. Morphological Traits and Mortality Rate

The diameter at the height of the neck (DHN) was measured with a digital meter foot and the height of the plant with a graduated ruler. The plant was dissected with a scalpel, and the root was cut from its first projection on the stem. Stems between the apical meristem and just before the first projection of the root were used. Seedlings were then dissected, separating the shoot and root organs, which were dried in an oven at 72 °C for 48 h, obtaining the dry weight (g). Seedling mortality was calculated as the frequency of seedlings that died at the end of the experiment (Figure 6).

### 4.13. Statistical Analysis

Data normality and variance homogeneity were assessed using the Shapiro–Wilk and Levene tests, respectively. Statistical significance was determined using Fisher’s LSD test (*p* < 0.05). A one-way ANOVA was performed to evaluate the effect of inoculation at the different irrigation levels (100, 75, and 25%) on seedlings. The effect of the interaction between inoculation and irrigation level was determined using factor analysis. For the analysis, the percentage values were transformed through Arcosin√(x/100). In addition, the data sets were subjected to principal component analysis (PCA) and hierarchical clustering. The farthest neighbor method was used to group the experimental individuals according to their associations with the PCA obtained. Statistical analyses were performed using the STATISTICA v.10 (Statsoft, Tulsa, OK, USA) and SPSS v.22.0 (IBM Corp. Armonk, NY, USA, EE. UU.) software packages.

## 5. Conclusions

The results obtained in this study show that inoculation with AMF and EF + AMF has positive effects on the survival rate of *A. araucana* under the most extreme drought conditions (25% FC). These conditions significantly improved tolerance to the stress produced by a water deficit. In contrast, the EF tested alone (in the absence of AMF) and under stressful abiotic conditions such as drought presented a null or neutral effect on *A. araucana* seedlings, with no improvement in water stress tolerance.

Overall, the results obtained here emphasize the importance of considering the microbiological component to develop *A. araucana* plants with the potential to adapt to adverse environmental conditions so that ecological restoration programs with this native species will be effective, mainly in the context of a water crisis as observed in the last 10 years in central–southern Chile, where *A. araucana* grows naturally.

## Figures and Tables

**Figure 1 plants-12-02116-f001:**
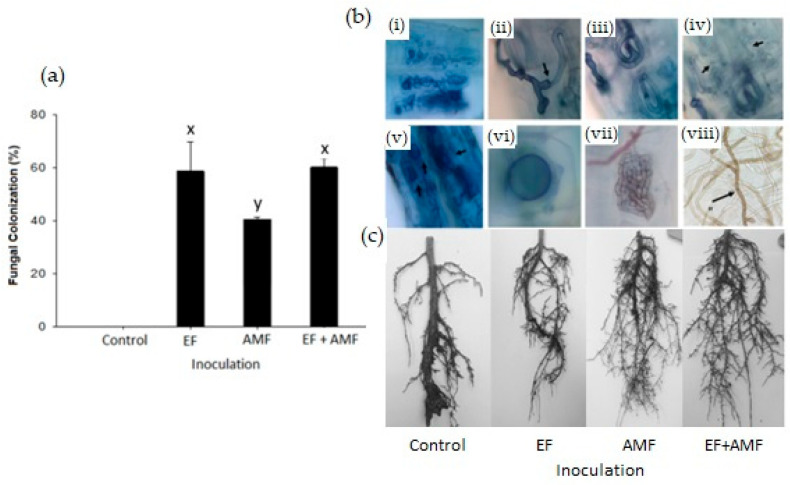
(**a**) Fungal colonization percentages in roots of *A. araucana* plants, including structures of AMF, EF, or the EF + AMF combination. (**b**) Diagnosis of AMF and EF structures in fine roots of *A. araucana*: (**i**) general view of an AMF colonizing *A. araucana* root; (**ii**) appressorium and entry point (arrowhead) on the root surface; (**iii**) hyphal coil; (**iv**) cell-cell connection (arrowhead); (**v**) irregularly shaped and ramified vesicles; (**vi**) globular vesicles; (**vii**) detail of microsclerotia of EF, (**viii**) dark septate of EF, and (**c**) root architecture development. The x and y letters in each bar in subfigure (**a**) indicate significant differences between treatments with *p* < 0.05 (ANOVA and Fisher’s LSD multiple-range test). Vertical lines on the bars indicate the standard deviation.

**Figure 2 plants-12-02116-f002:**
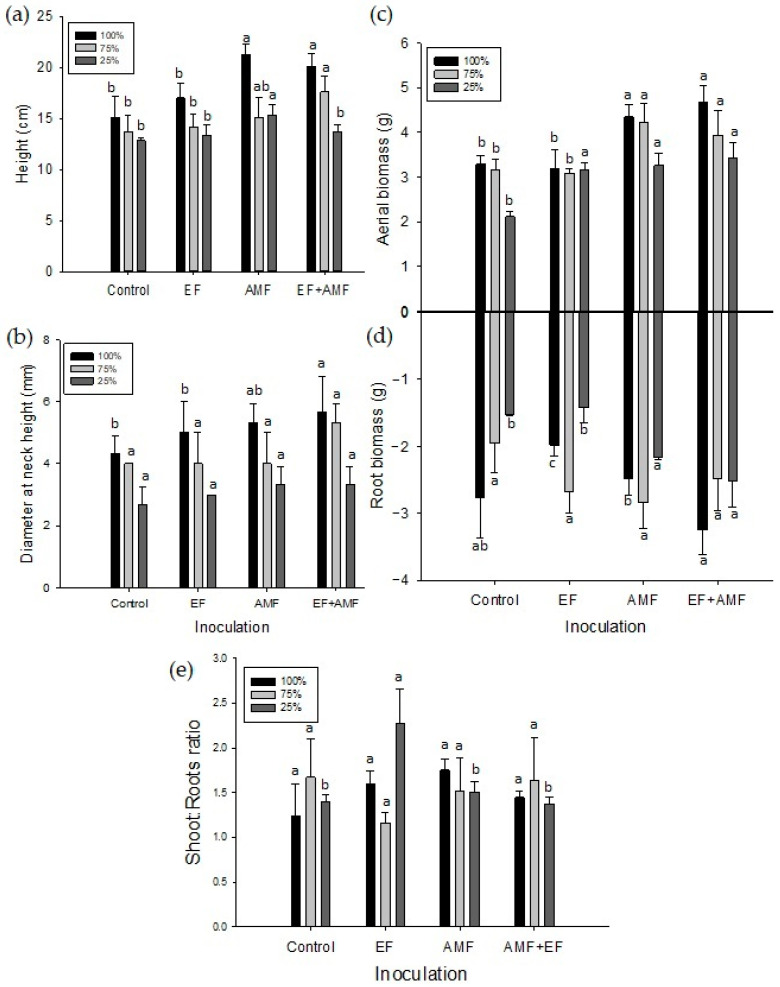
Morphological traits of *A. araucana* plants inoculated with AMF, EF, or EF + AMF and subjected to decreasing irrigation regimes. (**a**) Plant height; (**b**) diameter at the height of the neck; (**c**) aerial biomass; (**d**) root biomass; and (**e**) shoot:root ratio. Different letters in the bars indicate significant differences between treatments with *p* < 0.05 (ANOVA and Fisher’s LSD multiple-range test). Vertical lines on the bars indicate the standard deviation.

**Figure 3 plants-12-02116-f003:**
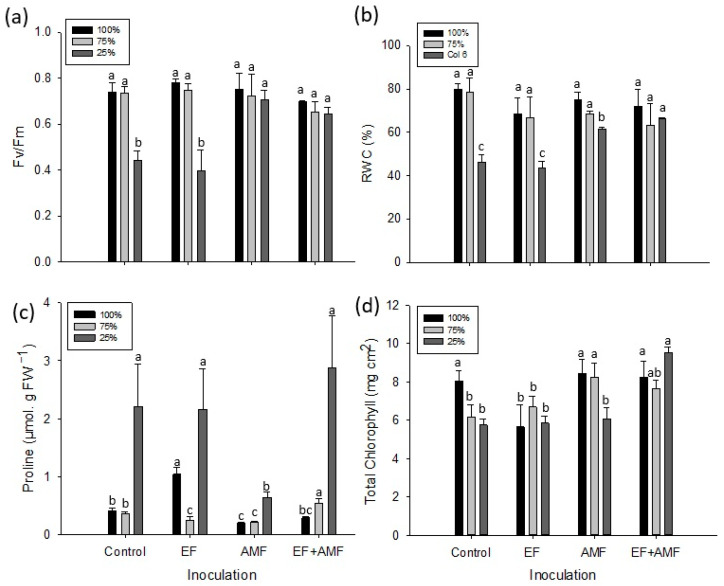
Physiological traits of *A. araucana* plants inoculated with AMF, EF, or EF + AMF and subjected to decreasing irrigation regimes. (**a**) Maximum quantum yield of photosystem II (Fv/Fm); (**b**) RWC; (**c**) proline content (μmol g FW^−^^1^); and (**d**) total chlorophyll (mg cm^2^). Different letters in the bars indicate significant differences among treatments with *p* < 0.05 (ANOVA and Fisher’s LSD multiple-range test), *n* = 3. Vertical lines on the bars indicate the standard deviation.

**Figure 4 plants-12-02116-f004:**
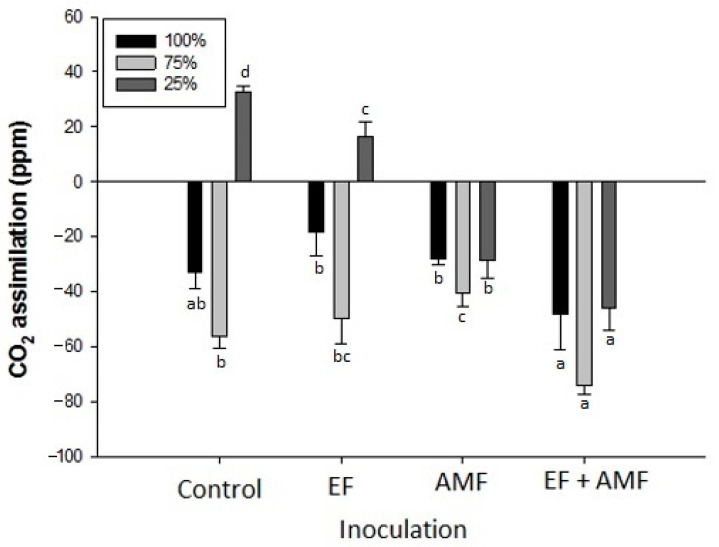
CO_2_ assimilation of *A. araucana* plants inoculated with AMF, EF, or EF + AMF and subjected to decreasing irrigation regimes. CO_2_ assimilation is shown in negative values and photorespiration in positive values. The measurement conditions were 1000 µmol m^−^^2^ s^−^^1^, a temperature of 22 °C, and a CO_2_ concentration of 400 µM. Different letters in the bars indicate significant differences among treatments with *p* < 0.05 (ANOVA and Fisher’s LSD multiple-range test), *n* = 3. Vertical lines on the bars indicate the standard deviation.

**Figure 5 plants-12-02116-f005:**
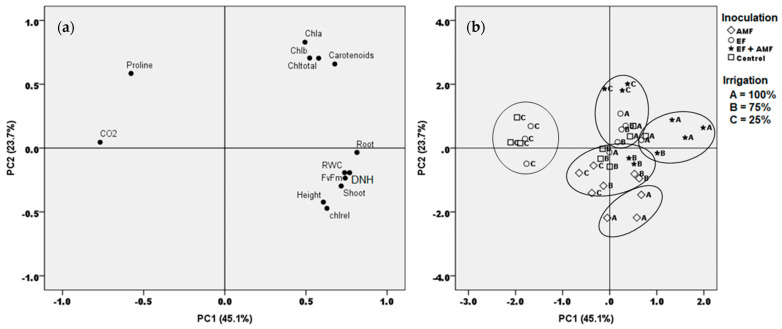
(**a**) Principal component (PC) scores for the experimental variables determined in seedlings of *A. araucana* plants inoculated with AMF, EF, or EF + AMF and subjected to decreasing irrigation regimes; and (**b**) grouping of the experimental individuals according to the different inoculation and irrigation treatments. The percentage values in parentheses indicate the variation explained by each PC. Abbreviations—CO_2_: CO_2_ assimilation; Chla: chlorophyll a; Chlb: chlorophyll b; Chltotal: total chlorophyll; Root: root dry biomass; Shoot: shoot dry biomass; RWC: relative water content; FvFm: maximum quantum yield of photosystem II; DHN: diameters at the height of the neck; chlrel: relative chlorophyll (SPAD values). The circles represent individuals of similar behavior according to the hierarchical cluster analysis using the farthest neighbor method.

**Figure 6 plants-12-02116-f006:**
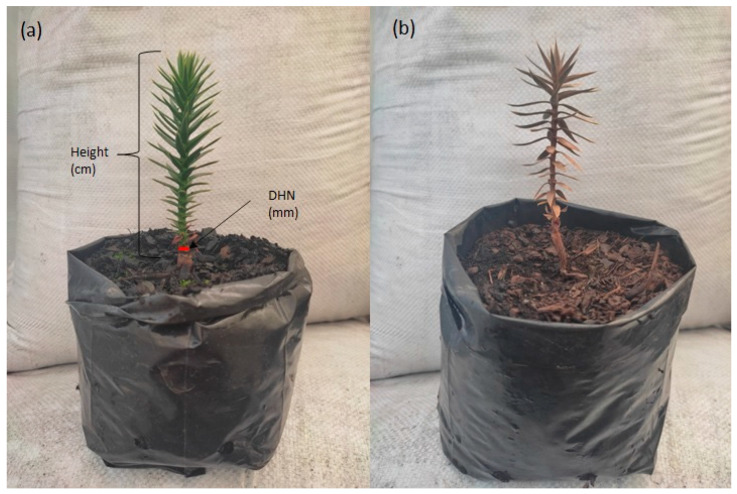
*Araucaria araucana* plants: (**a**) Considered a living plant with measurement of the diameter at the height of the neck (DNH); its height is indicated; (**b**) Considered a dead plant.

**Table 1 plants-12-02116-t001:** *F*-values and probabilities of significance for the main effects and factor interaction for the variables measured and analyzed by means of a two-way ANOVA in plants of *Araucaria araucana* inoculated with different types of fungi and growing at decreasing irrigation levels.

Sources of Variation	Experimental Variables								
	Height (cm)	DHN (mm)	Shoot (g)	Root (g)	S/R	Fv/Fm	RWC (%)	Proline (µmol g FW^−1^)	T-Chl (mg cm^−2^)
Inoculation	13.7 ***	3.70 *	29.4 ***	8.83 ***	1.41 ns	5.69 **	4.54 *	9.40 ***	23.4 ***
Irrigation	35.8 ***	25.1 ***	23.4 ***	14.4 ***	0.94 ns	50.0 ***	35.9 ***	63.4 ***	4.78 *
Inoc. X Irrig.	3.08 *	0.60 ns	3.41 *	4.31 **	4.94 **	11.6 ***	5.86 ***	5.67 ***	10.3 ***

Abbreviations: DHN = diameter at the height of the neck; S/R = shoot biomass to root biomass ratio; Fv/Fm = maximum quantum yield of photosystem II; RWC = relative water content; T-Chlorophyll = total chlorophyll content. Significance conventions: ns: not significant; * *p* < 0.05; ** *p* < 0.01; *** *p* < 0.001.

**Table 2 plants-12-02116-t002:** Mortality rates (%) for *A. araucana* seedlings under different irrigation conditions and fungal inocula.

Irrigation Levels	Control	EF	AMF	EF + AMF
100	0	0	0	0
75	0	0	0	0
25	75	87.5	50	25

## Data Availability

The data presented in this study are available on request from the corresponding author.

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
