# Peer review of "Contribution of Arbuscular Mycorrhizal and Endophytic Fungi to Drought Tolerance in Araucaria araucana Seedlings"

_plants, 2023, doi:10.3390/plants12112116_

Round 1
Reviewer 1 Report
In the present research article entitled “Contribution of arbuscular mycorrhizal and endophytic fungi to drought tolerance in Araucaria araucana seedlings” authors want to confirm the role of arbuscular mycorrhizal and endophytic fungi in improving the drought tolerance in Araucaria araucana seedlings. This study is routine and is based on lines of previous studies which have already shown the role of these inoculation treatments in stress alleviation, in particular water stress, in various plant species. In addition, this study do not contributes a new knowledge to plant physiology.
Actually, there are many flaws in this MS that it is not suitable for publication in Plants in its current form. It needs clarifications in many aspects after major revisions as follows:
The main concern is:
Abstract:
(1) The main results that were obtained to conclude their conclusion is not presented well in the abstract, presenting the results interesting to the readers. This part needs to be completely revised.
Introduction:
(2) With regards to arbuscular mycorrhizal and/or endophytic fungi, more attention should be paid on mechanisms of action of each of them, either separately or in combination, on regulation of stress tolerance especially drought tolerance. It lacks specific literature, previous and recent, on water stress.
(3) The introduction is written chaotically. There is no hypothesis or purpose of the study.
Results:
(4) The results need to interpreted by taking the effect of arbuscular mycorrhizal and endophytic fungi separately and then comparing their synergistic effects. This aspect is quite lacking in this paper. None of the results obtained when both arbuscular mycorrhizal and endophytic fungi are applied alone or in combination could confirm the reversal of water stress.
(5) Regarding AMF results in Table 2, I wonder that the group inoculated with AMF at 100% irrigation presented the lowest concentration of chlorophyll a, chlorophyll b, and carotenoids compared to the control treatment under the same irrigation condition.
However, as shown in Figure 2, the growth parameters were improved in the same group. Actually, decreasing photosynthetic pigments amount should reflected negatively on the plant growth and development.
(6) from the Figure 3, it was not convincing that inoculated plants grew better than the un-inoculated ones, the authors need to measure many other parameters such as photochemical reactions activity, gas exchange parameters (stomatal conductance, transpiration rate, intercellular CO2 concentration), chlorophyll fluorescence system [electron transport rate (ETR), actual photochemical efficiency of PSII (ΦPSII), photochemical quenching coefficient (qP), effective quantum yield of PSII photochemistry (Fv′/Fm'), and non-photochemical quenching coefficients (qN)], and photosynthetic enzymes activity to give their conclusion In line 26:
both the AMF and the dual inoculation allowed increased growth, photosynthesis.
Discussion:
(7) Discussion must be improved – the authors did not present any possible mechanisms of arbuscular mycorrhizal and/or endophytic fungi under both stressed and non-stressed conditions of each physiological/biochemical parameters. The mechanism of action of water stress as well as the effect of arbuscular mycorrhizal and/or endophytic fungi is missing in all parameters. Why and how water deficit condition increases or decreases a parameter and why and how arbuscular mycorrhizal and/or endophytic fungi increases or decreases a parameter……what are the possible mechanisms, should be mentioned in the discussion part.

Author Response
Reviewer 1:
In the present research article entitled “Contribution of arbuscular mycorrhizal and endophytic fungi to drought tolerance in Araucaria araucana seedlings” authors want to confirm the role of arbuscular mycorrhizal and endophytic fungi in improving the drought tolerance in Araucaria araucana seedlings. This study is routine and is based on lines of previous studies which have already shown the role of these inoculation treatments in stress alleviation, in particular water stress, in various plant species. In addition, this study do not contributes a new knowledge to plant physiology.
R: Thank you for the comment. We must highlight that in vivo studies using Araucaria araucana as host plant are very scarce, being mainly studied at ecosystem scale. Moreover, considering the several particularities of this species (extremely slow growth, limited distribution, high vulnerability to environmental factors and fire, among others) and the actual conservation state, the opportunity to generate new knowledge about sustainable alternatives to improve the establishment regarding ecological restoration is highly relevant. Especially, when alternatives to conditioning the (myco)rhizosphere environment are considered. Maybe the physiological new contribution of our work seems to be limited, but on the contrary, the other factor that involve the relation at the interface plant-microorganisms are very relevant, especially in this living fossil plant species, and practically unknown, which highlight the novelty of the results here reported.
Actually, there are many flaws in this MS that it is not suitable for publication in Plants in its current form. It needs clarifications in many aspects after major revisions as follows:
R: Dear colleague, we agree that our first version presented several flaws. However, a very complete and extensive improvement was performed. Please, see the changes (in red for an easy corroboration) throughout the manuscript.
The main concern is:
Abstract:
(1) The main results that were obtained to conclude their conclusion is not presented well in the abstract, presenting the results interesting to the readers. This part needs to be completely revised.
R: Done. Please, see the new version where interesting results supporting the conclusions were included, mainly in the second half of the section.
Introduction:
(2) With regards to arbuscular mycorrhizal and/or endophytic fungi, more attention should be paid on mechanisms of action of each of them, either separately or in combination, on regulation of stress tolerance especially drought tolerance. It lacks specific literature, previous and recent, on water stress.
R: Done. More antecedents were incorporated, mainly in the 3rd and 4th paragraph in the introduction. However, it must be clarified that specific antecedents regarding the role of the here used microorganisms at mechanistic level under drought conditions are practically non-existent.
(3) The introduction is written chaotically. There is no hypothesis or purpose of the study.
R: Dear reviewer, as you can see, the new version of Introduction section is most structured, according to your previous comments, including a well-stated purpose for this study.
Results:
(4) The results need to interpreted by taking the effect of arbuscular mycorrhizal and endophytic fungi separately and then comparing their synergistic effects. This aspect is quite lacking in this paper. None of the results obtained when both arbuscular mycorrhizal and endophytic fungi are applied alone or in combination could confirm the reversal of water stress.
R: Done. As you can see, and according with other comments (questions 5 and 6), and some observations of Reviewer 2 we re-do the analysis. Effectively, some incongruences were detected due to confusion in the representation of some results, changing the values in the treatments with inoculation. Therefore, we carefully revised all the data, taking the opportunity to obtain the significance for the different experimental factors and interaction, but as it is possible to observe (Table 2) practically all the experimental variables were significantly influenced by the factor interaction. Anyway, we were reporting the results (and discussion) based also in the tendencies attributable to some of the individual factors and levels.
(5) Regarding AMF results in Table 2, I wonder that the group inoculated with AMF at 100% irrigation presented the lowest concentration of chlorophyll a, chlorophyll b, and carotenoids compared to the control treatment under the same irrigation condition.
However, as shown in Figure 2, the growth parameters were improved in the same group. Actually, decreasing photosynthetic pigments amount should reflected negatively on the plant growth and development.
R: We partially agree. Maybe the total content of pigments could represent a more direct relationship between both variables. However, as observed in fig. 5 this relationship (not strong) is present, but seems to be more determinant the water content, which is directly influenced by symbiotic fungi, mainly AM fungi. In the same sense with your next questions, some other variables related to photosynthesis are needed to explore, but with the available equipment at our lab this is not possible yet. Please, see the next explanation.
(6) from the Figure 3, it was not convincing that inoculated plants grew better than the un-inoculated ones, the authors need to measure many other parameters such as photochemical reactions activity, gas exchange parameters (stomatal conductance, transpiration rate, intercellular CO2 concentration), chlorophyll fluorescence system [electron transport rate (ETR), actual photochemical efficiency of PSII (ΦPSII), photochemical quenching coefficient (qP), effective quantum yield of PSII photochemistry (Fv′/Fm'), and non-photochemical quenching coefficients (qN)], and photosynthetic enzymes activity to give their conclusion In line 26:
both the AMF and the dual inoculation allowed increased growth, photosynthesis.
R: We agree that other traits are relevant for convincingly support our results and conclusions. In fact, we usually use practically all the experimental variables that you mentioned in (6). For this, we use two equipment, one of them based in gas exchange (Targas-1) and the other based in fluorescence parameters (FluorPen). Unfortunately, the structure of the Araucaria plantlets and needles make impossible to use the chambers to measure the gas exchange, since the needles are very gross, coriaceous, and directly attached to the stem, so it was not possible to take a lecture without damage the entire plantlet (see figure 6 for a visual idea about the morphology of the plantlets). This is the reason why we modified a system consisting of a chamber to collect and measure CO2 from soil coupled to the equipment, which allow us to measure at least the CO2 exchange starting of environmental concentrations and after homogeneous time for all the plantlets (into bags excluding the soil emissions). We hope to search for and acquire some other accessories or equipment in the near future for a more accurate measure of photosynthetic parameters that allow us the use in this plant species or other with similar particularities.
Discussion:
(7) Discussion must be improved – the authors did not present any possible mechanisms of arbuscular mycorrhizal and/or endophytic fungi under both stressed and non-stressed conditions of each physiological/biochemical parameters. The mechanism of action of water stress as well as the effect of arbuscular mycorrhizal and/or endophytic fungi is missing in all parameters. Why and how water deficit condition increases or decreases a parameter and why and how arbuscular mycorrhizal and/or endophytic fungi increases or decreases a parameter……what are the possible mechanisms, should be mentioned in the discussion part.
R: Done. As you can see, the mechanistic regarding the presence of fungi (at least more known for AM fungi) was incorporated in the first half of the section, also including the relationship that we observed regarding decreases and increases in the here analyzed variables. Again, thank you for your deep and complete comments and questions. We feel that our manuscript (including an English revision) has substantially increased the quality.
Reviewer 2 Report
In general, this study was designed well, conducted well, and abundant data was collected. However, the data was not analyzed in a right way. The majority of the results and discussion needs to be re-wrote after correcting the statistical analysis method. English language needs to be improved. Discussion needs to be improved a lot.
Major comments
Figures 2&3, table 2: The mean separation should be done by different irrigation method (100%, 75%, and 25%) instead of comparing all the treatment combinations at one time. For example, in Figure 2(a), under 75% irrigation method, ‘b b b ab’ doesn’t make any sense, there is no significance among the 4 inoculations, it should be ‘a a a a’ instead of ‘b b b ab’. However, if authors run the right mean separation, it is possible there is significant difference between inoculation treatments under the same irrigation method. Another statistical analyses way is, Two-way ANOVA. Since authors mentioned factorial ANOVA in lines 582-583, the interactions between inoculation and irrigation methods should be studied first. If the interaction is no significant, then look at the main effects.
Discussion: There are 25 paragraphs. Can authors please reorganize the logic flow in the discussion to make it not too scattered?
Lines 292-304: These 3 paragraphs should be included in introduction instead of discussion.
Minor comments
Lines 439-441: Do authors mean soil portion collected in 125, 45, 32 µm mesh? The description is confusing here.
Line 474: What is the meaning of ‘c.a.’?
Line 475-477: I am curious about the isolation results of the endophytic fungi, did you get some Petri dishes with full of molds? If yes, will this affect your results (since they will affect the isolation of other fungi)?
Line 506: typo
Line 518: ‘inoculation’ is not accurate here, inoculation should be finished after the injection/application of fungi.
Line 522: plants
Line 570: What do authors mean by ‘the diameter at the height of the neck’? If possible, can authors show the definition of this parameter in a picture like figure 6a?
Figure 5a: What do authors mean by ‘chlrel’?
Lines 301-304: Are these beneficial effects of EF come from the same species as mentioned in lines 491-492?
Line 323: ‘was not enough time’ is not proper here, please re-write.
Lines 343-345: English language needs to be improved; I don’t understand what authors want to express here.
Lines 418-424: The first sentence in this paragraph let me expect a summary of the findings in study. However, authors summarized results from other studies. Again, this should be included in the introduction instead of discussion.
Author Response
Reviewer 2
In general, this study was designed well, conducted well, and abundant data was collected. However, the data was not analyzed in a right way. The majority of the results and discussion needs to be re-wrote after correcting the statistical analysis method. English language needs to be improved. Discussion needs to be improved a lot.
R: Dear colleague, thank you for your positive comments regarding our study. As you can see, all the comments were incorporated, including new statistics and re-doing the figures, results and discussion accordingly.
Major comments
Figures 2&3, table 2: The mean separation should be done by different irrigation method (100%, 75%, and 25%) instead of comparing all the treatment combinations at one time. For example, in Figure 2(a), under 75% irrigation method, ‘b b b ab’ doesn’t make any sense, there is no significance among the 4 inoculations, it should be ‘a a a a’ instead of ‘b b b ab’. However, if authors run the right mean separation, it is possible there is significant difference between inoculation treatments under the same irrigation method. Another statistical analyses way is, Two-way ANOVA. Since authors mentioned factorial ANOVA in lines 582-583, the interactions between inoculation and irrigation methods should be studied first. If the interaction is no significant, then look at the main effects.
R: Done, please excuse us the confusion. There were some mistakes in the previous version consisting in the change of the inoculated treatments in the figure, as was also mentioned to the reviewer 1. Please, see also the new table 2. The previous table 2 was taken off, since the results are redundant, because chla + chlb is the total chlorophyll (showed in fig. 3d) strongly correlated (fig. 5a).
Discussion: There are 25 paragraphs. Can authors please reorganize the logic flow in the discussion to make it not too scattered?
R: Done, we make deep changes throughout the entire manuscript. Detailing, the discussion section was reordered, and we feel that the new version is more concise and organized in a flow similar to the other sections, and in only 10 paragraphs.
Lines 292-304: These 3 paragraphs should be included in introduction instead of discussion.
R: we agree with your suggestion. Please, corroborate the changes at the beginning of the discussion section and in the second and following paragraphs in the introduction section.
Minor comments
Lines 439-441: Do authors mean soil portion collected in 125, 45, 32 µm mesh? The description is confusing here.
R: It was a mistake. Actually, the finest fractions are recovered. Please, see the new version at lines 408-411.
Line 474: What is the meaning of ‘c.a.’?
R: Thank you for the comment. We changed to “pieces of 5 mm” (line 442).
Line 475-477: I am curious about the isolation results of the endophytic fungi, did you get some Petri dishes with full of molds? If yes, will this affect your results (since they will affect the isolation of other fungi)?
R: Effectively, during the isolation some dishes were colonized by molds, even more than one fungus. However, according to the laboratory biosecurity protocols, the dishes that not produced a unique species were discarded before the molecular identification.
Line 506: typo
R: Done. Changed to “Seedlings of A. araucana were left for 5 months for root colonization” (L472-473).
Line 518: ‘inoculation’ is not accurate here, inoculation should be finished after the injection/application of fungi.
R: Done. Changed to “post-inoculation” (485).
Line 522: plants
R: Done. Plural “plants” (489).
Line 570: What do authors mean by ‘the diameter at the height of the neck’? If possible, can authors show the definition of this parameter in a picture like figure 6a?
R: Done. Please, see the details in the new Fig. 6.
Figure 5a: What do authors mean by ‘chlrel’?
R: Done, please the caption of the fig. 5, “chlrel: relative chlorophyll (SPAD values)”.
Lines 301-304: Are these beneficial effects of EF come from the same species as mentioned in lines 491-492?
R: The changes performed included deep changes in this section, which are more useful at the introduction, as suggested in a previous comment.
Line 323: ‘was not enough time’ is not proper here, please re-write.
R: Done, the sentence was changed accordingly. “According to the results obtained in this work, the 7-month duration of the study (5 months of colonization plus 2 months of water stress) allowed for substantial significant morphophysiological changes in the groups of plants subjected to drought stress” (L281-283).
Lines 343-345: English language needs to be improved; I don’t understand what authors want to express here.
R: Done. Some mistakes were included in the previous version. “Studies in this regard have been carried out only for grasses, where xerotolerance has been obtained via symbiosis using EF from the Clavicipitaceous group, fungi that can only colonize grasses and not woody species” (L320-323).
Lines 418-424: The first sentence in this paragraph let me expect a summary of the findings in study. However, authors summarized results from other studies. Again, this should be included in the introduction instead of discussion.
R: Done. We circumscribe the paragraph mainly to our results and the porjections in this new version. “The results presented in our study suggest that the combination of these micro-organisms (AMF and EF) was beneficial for Araucaria araucana seedlings, since EF stimulate proline synthesis [68] and AMF attenuate the possible adverse effects of en-dophytes under abiotic stress [69,70,25]. Moreover, fungal inocula can promote high percentages of water in the leaves, notably improving tolerance to drought stress [71]. These results suggest the importance of the combination of symbiotic microorganisms in both the rhizosphere and in plant organs, improving tolerance to abiotic stress” (L389-395).
Dear reviewer, thank you again for your valuable contribution.
Round 2
Reviewer 1 Report
Some comments have been addressed, however the parameters which measured by authors is not enough to discuss their title. To be honest I also expected to see other data, which may be on the macro- and micro- nutrients, the organic solutes accumulation, the photosynthetic activity, the protein synthesis, the reactive oxygen species scavenging system (enzymatic and non-enzymatic detoxification systems to counteract ROS),……………….. to get a better inside into the response of Araucaria araucana seedlings to arbuscular mycorrhizal and endophytic fungi under drought conditions. So, I am asking authors to do extra work by measuring more parameters and resubmit their manuscript again.
Author Response
Reviewer 1.
Some comments have been addressed, however the parameters which measured by authors is not enough to discuss their title. To be honest I also expected to see other data, which may be on the macro- and micro- nutrients, the organic solutes accumulation, the photosynthetic activity, the protein synthesis, the reactive oxygen species scavenging system (enzymatic and non-enzymatic detoxification systems to counteract ROS),……………….. to get a better inside into the response of Araucaria araucana seedlings to arbuscular mycorrhizal and endophytic fungi under drought conditions. So, I am asking authors to do extra work by measuring more parameters and resubmit their manuscript again.
R: Dear reviewer, we would also have liked to show an extensive battery of data of various other experimental variables, not only those you mention, probably even including integration of omics tools. The above can surely answer complete scientific questions on the subject; however, it is also necessary to first partially know the responses of a species with its particularities, especially Araucaria, which has several. We do not agree with you that with the data presented it is not possible to discuss the title. This is broad, and not limited to physiological aspects, such as those requested by you in the first round, or to biochemical aspects, such as the requested in this second round. In fact, it was established according to the objective, as previously requested, in morphophysiological aspects, with variables that could be feasible to measure in the plant species studied. Again, we want to clarify that this study is one of the first (maybe the unique) in this species considering exogenous management of symbiotic microorganisms, which is really valuable because of the symbiotic dependence of Araucaria, which at the moment has only been studied as adults, at the macro scale in their ecosystems. In addition, as previously mentioned in R1, this species presents an extremely slow growth; therefore, being able to meet the requirements implies carrying out the study again, with seeds that will be available only depending on the season, very slow germinations, to only after that wait another time during five months as explained in the design, to have plant material that probably does not reach for all the variables mentioned before in R1 and now in R2. Again, the morphology of the species does not allow the use of equipment that in other species is very easy to use (for example, gas exchange by means of IR), reason why we also previously included some images of the individuals that we hoped would also serve to demonstrate it. Detailing, the needles are impossible to place in any of the available cameras, and if there are accessories for this they will depend on the equipment (at least for Targas-1 PP systems, which is the one we have in our institution, does not exist), or alternatively investments of some tens of thousands USD for a new equipment. In addition, consider that this trial was carried out during a long quarantine in Chile, without greater access to collaborators' equipment or other resources that would allow greater layers of experimental variables. In this scenario, I also highlight the successful obtaining of these novel results. The data are probably limited; however, they do not seek to be final results on the role of symbiont fungi in Araucaria, but they are certainly important as an advance and an excellent preliminary support to the important role of them in tolerance to water stress. Finally, I must emphasize that being able to comply with your request means a delay of at least one year for a new study, which is materially impossible. Under these limitations, if you consider the study and these results are not appropriate for publication, please recommend in that sense, and if the editor considers your argument valid, we can obtain a fast reject of our manuscript for continuing the process in another journal.
Best regards
Reviewer 2 Report
The y-axis name/title of figure 4 needs to be improved, since 'ppm' should be the unit, and this figure presented CO2 assimilation of plants.
Table 2 should be inserted between 2.1 and 2.2, instead of being included in 2.6.
For other sections, I am pleased to report that the authors have responded to all of my comments in a reasonable and satisfactory manner.
Author Response
Reviewer 2.
The y-axis name/title of figure 4 needs to be improved, since 'ppm' should be the unit, and this figure presented CO2 assimilation of plants.
R: Effectively is only the unit and no the variable, we agree with you. Done, please, see the figure.
Table 2 should be inserted between 2.1 and 2.2, instead of being included in 2.6.
R: Done. Moreover, we have moved the figure 2 after the first paragraph mentioning the figure and checked that all the other aspects are correct.
For other sections, I am pleased to report that the authors have responded to all of my comments in a reasonable and satisfactory manner.
R: Thank you for your support and valuable comments about our study.